# DuoFormer: Leveraging Hierarchical Representations by Local and Global Attention Vision Transformer

**Xiaoya Tang**[1,2]                                XIAOYA.TANG@UTAH.EDU
**Bodong Zhang**[1,3]                          BODONG.ZHANG@UTAH.EDU
**Man Minh Ho**[1]                           MANMINHHO.CS@GMAIL.COM
**Beatrice S. Knudsen**[4]               BEATRICE.KNUDSEN@PATH.UTAH.EDU
**Tolga Tasdizen**[1,3]                          TOLGA@SCI.UTAH.EDU

[1] *Scientific Computing and Imaging Institute, University of Utah, SLC, UT, USA*

[2] *Kahlert School of Computing, University of Utah, SLC, UT, USA*

[3] *Electrical and Computer Engineering, University of Utah, SLC, UT, USA*

[4] *Department of Pathology, University of Utah, Salt Lake City, UT, USA*

**Editors:** Accepted for publication at MIDL 2025

## Abstract

Despite the widespread adoption of transformers in medical applications, the exploration of multi-scale learning through transformers remains limited, while hierarchical representations are considered advantageous for computer-aided medical diagnosis. We propose a novel hierarchical transformer model that adeptly integrates the feature extraction capabilities of Convolutional Neural Networks (CNNs) with the advanced representational potential of Vision Transformers (ViTs). Addressing the lack of inductive biases and dependence on extensive training datasets in ViTs, our model employs a CNN backbone to generate hierarchical visual representations. These representations are adapted for transformer input through an innovative patch tokenization process, preserving the inherited multi-scale inductive biases. We also introduce a scale-wise attention mechanism that directly captures intra-scale and inter-scale associations. This mechanism complements patch-wise attention by enhancing spatial understanding and preserving global perception, which we refer to as local and global attention, respectively. Our model significantly outperforms baseline models in terms of classification accuracy, demonstrating its efficiency in bridging the gap between Convolutional Neural Networks (CNNs) and Vision Transformers (ViTs). The components are designed as plug-and-play for different CNN architectures and can be adapted for multiple applications. The code is available at https://github.com/xiaoyatang/DuoFormer.git.

**Keywords:** Vision Transformer, Inductive Bias, Multi-scale features

## 1. Introduction

The Vision Transformer (ViT) (Dosovitskiy et al., 2020) adapted transformers from language to vision, demonstrating superior performance over CNNs when pre-trained on large datasets. ViT employs a patch tokenization process that converts images into a sequence of uniform token embeddings. These tokens undergo Multi-Head Self-Attention (MSA), transforming them into queries, keys, and values that capture extensive non-local relationships. Despite their potential, ViTs can underperform similarly-sized ResNets (He et al., 2016) when inadequately trained due to their lack of inductive biases such as translation

equivariance and locality(Lee et al., 2021), which are naturally encoded by CNNs. Recent efforts have focused on mitigating ViTs' limitations by integrating convolutions or adding self-supervised tasks (Liu et al., 2021a). Prevalent approaches combine CNN feature extractors with transformer encoders (Araujo et al., 2019; Wu et al., 2021; Yuan et al., 2021; Li et al., 2021; d'Ascoli et al., 2021; Zhang and Yan, 2023; Hou et al., 2024), such as the 'hybrid' ViT (Dosovitskiy et al., 2020). Other methods such as knowledge distillation (Touvron et al., 2021) transfer biases from CNNs to ViT, add a convolutional kernel to the attention matrix to bring translation equivariance (Dai et al., 2021), and use pooling to build multi-stage transformers (Li et al., 2022). Nonetheless, ViTs' uniform representations throughout layers and their non-local receptive fields compared to CNNs limit their ability to capture detailed semantics (Raghu et al., 2021), which is important for medical images.

The application of ViTs in medical imaging, particularly in CT and X-ray data, is gaining momentum, showcasing their potential in handling extensive datasets (Shamshad et al., 2023). A notable application in histopathology is presented by (Shao et al., 2021), which utilizes transformers to understand correlations between patches in whole slide images (WSIs), demonstrating the adaptability of transformers for complex pathological data. Histopathology image analysis involves examining WSIs to detect and interpret complex tissue structures and cellular details. This analysis faces challenges due to similar appearances between background and tumor areas, as well as the varied scales of visual entities within WSIs. These include differences in sizes of cell nuclei and vascular structures, both of which can significantly impact a model's ability to differentiate between low- and high-risk kidney cancers as an example. Moreover, global features of cancer and its microenvironment, observable only at lower scales, are crucial for various downstream tasks. The neglect of these multiple scales can significantly impair the performance of deep learning models in medical image recognition tasks. CNNs tackle this issue by utilizing a hierarchical structure created by lower and higher stages. Such hierarchical structures are thought to be advantageous for cancer diagnosis and prognosis tasks. However, CNNs fall short in extracting the global contextual information crucial for medical image classification compared to Transformers. By harnessing a hierarchical structure similar to that of CNNs, ViTs can be prevented from overlooking the critical multi-scale features, while also imparting necessary inductive biases. Most existing works on directly integrating multi-scale information into ViTs vary primarily in the placement of convolutional operations: during patch tokenization(Yuan et al., 2021; Xu et al., 2021; Guo et al., 2022), within(Guo et al., 2022; Lin et al., 2023; Fan et al., 2024) or between self-attention layers, including query/key/value projections(Wu et al., 2021; Yuan et al., 2021), forward layers (Li et al., 2021), or positional encoding (Xu et al., 2021), etc. Recent advancements, such as those by (Liu et al., 2024), which leverage a feature pyramid and a k-NN graph to enhance local feature representation in histopathological images, reflect a growing trend in adopting hierarchical architectures tailored for medical datasets (Azad et al., 2024). Inspired by the Swin Transformer (Liu et al., 2021b), a shifting window strategy bringing locality to transformer, Chowdary and Yin (2024) used different window sizes in attention mechanisms and shifted window blocks to improve the accuracy of thoracic disease classification. Manzari et al. (2023) employed both convolutions and poolings before and inside the attentions for medical data classification. Luo et al. (2022) fused a UNet and a transformer, employing two cross-attention modules to enhance medical image segmentation. Wang et al. (2022) proposed a channel attention to bridge the

semantic gap between different stages of a UNet on medical image segmentation. Pina et al. (2024) applied a multi-scale deformable transformer (Zhu et al., 2020) to cell detection and classification. Additionally, Guo et al. (2023) considered a WSI pyramid as a hierarchical graph and employed a hierarchical graph-transformer to communicate between different resolutions of the WSI pyramids, thus improving the analysis of these images. Despite the benefits of hierarchical configurations, a definitive model for medical image analysis has not yet been established. Challenges persist in effectively producing and utilizing features across various scales, with the influence of different scales requiring further exploration.

To address these challenges, we propose a novel hierarchical Vision Transformer model. First, our proposed multi-scale tokenization involves a single-layer projection, patch indexing, and concatenation, assembling features from different stages of the CNN into multi-scale tokens, facilitating a richer representation of an image. Second, we introduce a novel local attention mechanism, combined with global patch attention, enabling the model to learn connections between scales. This approach effectively bridges the gap between CNN and Transformer architectures and various scales of features. Finally, our proposed scale token, part of the scale attention, is initialized with a fused embedding derived from hierarchical representations. It enriches the transformer's multi-granularity representation and aggregates scale information, serving as the input for the global patch attention.

## 2. Methodology

### 2.1. Multi-scale Patch Tokenization

The pipeline of our model is depicted in Figure 1. We replaced the embedding layer commonly used in ViTs with a pretrained CNN backbone, which produces hierarchical features with decreasing spatial resolutions and increasing channel dimensions. We introduced a novel patch tokenization process to adapt these hierarchical features for the transformer. This process extracts features from different stages and performs embeddings based on them using single-layer projections. Given the input of an image, $\mathbf{x} \in \mathbb{R}^{H \times W \times 3}$ with $H = W$, we derive hierarchical outputs from multiple stages, denoted as $\mathbf{x}_i \in \mathbb{R}^{P_i \times P_i \times C_i}$ for $i \in \{0, 1, 2, 3\}$. Here, $i$ denotes the $i^{\text{th}}$ stage in the CNN backbone, where $P_i = \frac{H}{4 \cdot 2^i}$ specifies the spatial resolution, and $C_i$ indicates the channel dimension. We then apply a linear projection to transform all the features into embeddings with dimension $D$. We refer to the subsequent embeddings as multi-scale embeddings, denoted by $\mathbf{x}'_i$, where $\mathbf{x}'_i \in \mathbb{R}^{P_i \times P_i \times D}$, as formulated in Equation (1).

$$\mathbf{x}'_i = \text{Projection}(\mathbf{x}_i) \tag{1}$$

Next, we split the multi-scale embeddings $\mathbf{x}'$ into $N$ non-overlapping patches, and flatten the spatial dimensions of them. Thus, for each scale, we obtain a sequence of tokens, each token with a spatial size $P'^2_i$, where $P'_i = \frac{H}{4 \cdot 2^i \cdot \sqrt{N}}$, for $i \in \{0, 1, 2, 3\}$. We index and concatenate the corresponding tokens across multiple scales for each patch to form the multi-scale tokens $\mathbf{X}^t_{\sum}$. This process is explained by Equations (2) and (3).

$$\mathbf{x}''_i \in \mathbb{R}^{N \times P'^2_i \times D} \tag{2}$$

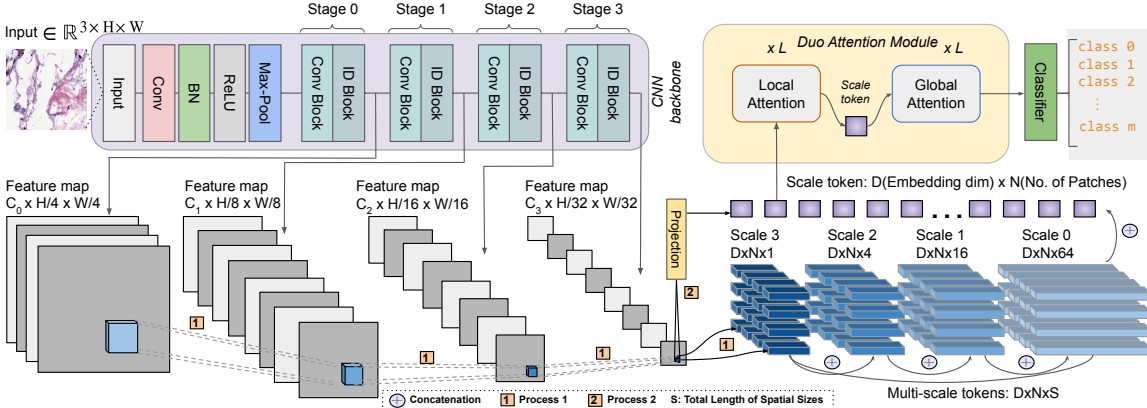

Figure 1: Left: Converting an image into hierarchical representations using a CNN backbone. Right: Process 1 illustrates multi-scale patch tokenization, including projection, patch splitting, and concatenation, with blue cubes representing embeddings from multiple scales of a single patch. Various colors and lengths indicate different embedding lengths at each scale in the multi-scale tokens. 'S' denotes the total embedding lengths for each patch. Process 2 shows learning the scale token from hierarchical representations. L indicates the depth of attention modules.

, where $P_i'^2 = \frac{HW}{16 \cdot 4^i \cdot N}, i \in \{0, 1, 2, 3\}$. In Equation (3), $S$ denotes the total embedding length of each embedded patch in the multi-scale tokens, and $S = \sum P_i'^2$.

$$\mathbf{X}_{\sum}^t = \mathbf{concat}(\mathbf{x}_i'') \in \mathbb{R}^{N \times S \times D} \tag{3}$$

## 2.2. Scale Token

In local attention for scale, a scale token—akin to the class token—aggregates scale information and is then passed into the global attention. We obtained the scale token $\mathbf{x}_s$ by applying a downsampling strategy to the hierarchical representations from the CNN, explained in Equation (4). This strategy normalizes the spatial dimensions of embeddings from different scales to $N$, maintaining consistent channel dimensions. $N$ denotes the number of patches. These embeddings are then concatenated along the channel dimension and projected into a dimension $D$ using a simple projection, illustrated by process 2 in Figure 2 and outlined in Equation (5). The resultant scale token distills important multi-scale information, serves as an effective guide for the local attention and efficiently aggregates scale information.

$$\tilde{\mathbf{x}}_0 = \mathrm{MaxPool}(\mathrm{Conv}(\mathbf{x}_0)), \tilde{\mathbf{x}}_1 = \mathrm{MaxPool}(\mathrm{Conv}(\mathbf{x}_1)),$$
$$\tilde{\mathbf{x}}_2 = \mathrm{MaxPool}(\mathbf{x}_2), \tilde{\mathbf{x}}_3 = \mathbf{x}_3, \quad \text{where } \mathbf{x}_i \in \mathbb{R}^{N \times C_i}, \tag{4}$$
$$\tilde{\mathbf{X}}_{\sum} = \mathbf{concat}(\tilde{\mathbf{x}}_0, \tilde{\mathbf{x}}_1, \tilde{\mathbf{x}}_2, \tilde{\mathbf{x}}_3) \in \mathbb{R}^{N \times C}, C = \sum C_i,$$

$$\mathbf{x}_s = \mathrm{ReLU}(\mathrm{BN}(\mathrm{Conv}(\tilde{\mathbf{X}}_{\sum}))) \in \mathbb{R}^{N \times D} \tag{5}$$

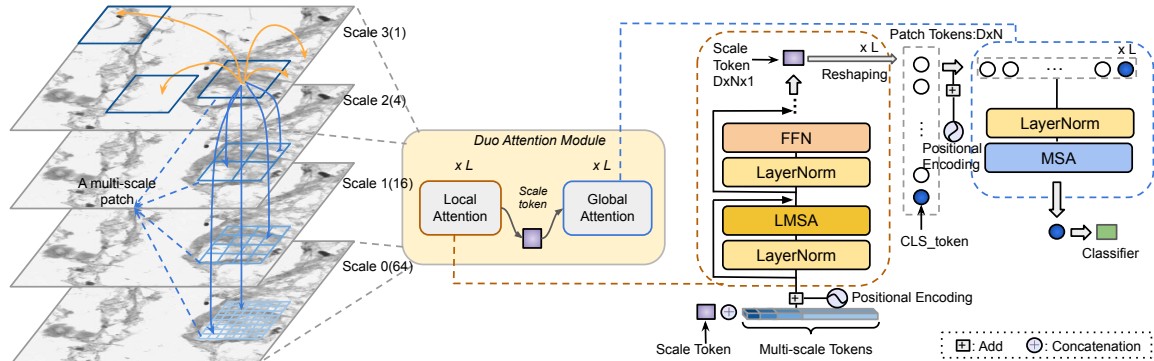

Figure 2: Left: Local (blue arrows) attention models intra- and inter-scale dependencies, while global (orange arrows) attention models relationships among image patches. From top to bottom, the embedding length (spatial sizes) for a single patch increases from 1 to 64, with rich scale information embedded in the multi-scale tokens. Right: Implementation of the duo attention module, including L layers of local and global attentions, respectively.

## 2.3. Duo Attention Module

Our encoder employs local and global attentions to respectively focus on detailed image features and broader contexts, as illustrated in Figure 2(Left). The local attention(LMSA) follows the principles of Multi-Head Self-Attention (MSA) but includes an adaption to incorporate an additional scale dimension. This adaptation integrates multi-scale analysis directly into the attention mechanism and modifies tensor operations to accommodate multi-scale tokens. Learnable 2D positional embeddings are added before the first layer of local attention, encoding scale-wise information for every patch. The implementations are depicted in Figure 2(Right), taking the first layer as an example. The input to the first local attention layer is denoted by $\mathbf{X}_0^s$ in Equation (6). $\mathbf{W}_{qkv}$ is transformation matrix. Equation (6) details the calculations performed within a local attention(LMSA) in a single head. $d_k$ is the scaling factor, and $\mathbf{A}$ stands for the attention weights for scales. We use multi-head attention in implementation.

$$
\begin{aligned}
\mathbf{X}_0^s &= \mathbf{concat}(\mathbf{x}_s, \mathbf{X}_{\sum}^t) + \mathbf{E}_{pos}, \mathbf{E}_{pos} \in \mathbb{R}^{(S+1) \times D}, \\
\mathbf{X}_0^{s'} &= \mathbf{X}_0^s + \mathrm{LMSA}(\mathrm{LN}(\mathbf{X}_0^s)), \mathbf{Y}_0^s = \mathbf{X}_0^{s'} + \mathrm{FFN}(\mathrm{LN}(\mathbf{X}_0^{s'}) \\
[\mathbf{qkv}] &= \mathbf{X}_0^s \mathbf{W}_{qkv}, \mathbf{W}_{qkv} \in \mathbb{R}^{D \times 3D}, \mathbf{q}/\mathbf{k}/\mathbf{v} \in \mathbb{R}^{N \times (S+1) \times D}, \\
\mathbf{A} &= \mathrm{Softmax}\left(\frac{\mathbf{q}\mathbf{k}^T}{\sqrt{d_k}} \mathbf{V}\right), d_k = \frac{D}{n_h}, \mathbf{A} \in \mathbb{R}^{(S+1) \times (S+1)},
\end{aligned}
\tag{6}
$$

After the local scale-wise attention, each patch is expected to encapsulate the necessary details across all scales. The scale token, aggregating key details from this module, is augmented with standard learnable 1D positional embeddings and passed into the global patch attention. Global attention mirrors standard MSA but empirically removes layer normalization(LN), feed-forward networks (FFN), and residual connections, as demonstrated in Figure 2(Right). A classifier, consisting of a single linear layer, is attached to the $L_{th}$ global attention layer, taking the CLS_token as the final image representation.

## 3. Experiments

### 3.1. Experimental Setup

Our evaluation utilized two datasets, Utah ccRCC and TCGA ccRCC (Zhang et al., 2023). The Utah ccRCC dataset comprises 49 WSIs from 49 patients, split into training (32 WSIs), validation (10 WSIs), and testing (7 WSIs). Tiles were extracted from marked polygons at 400x400 pixel resolution at 10X magnification with a 200-pixel stride and center-cropped to 224x224 pixels for model compatibility. The training set included 28,497 Normal/Benign, 2,044 Low Risk, 2,522 High Risk, and 4,115 Necrosis tiles, with validation and test sets proportionately distributed. The TCGA ccRCC dataset features 150 labeled WSIs divided into 30 for training, 60 for validation, and 60 for testing, using similar cropping methods but adjusted strides to gather more training patches. It contains 84,578 Normal/Benign, 180,471 Cancer, and 7,932 Necrosis tiles in the training set, with similar distributions in the validation and test sets.

All models, including baselines, were trained using the Adam optimizer with $\beta_1 = 0.9$ and $\beta_2 = 0.999$, without applying weight decay. For the DuoFormer model, batch sizes were set to 32 for the Utah dataset and 6 for the TCGA dataset. We employed a OneCycle learning rate scheduler that starts from a minimal learning rate, progressively increasing to a set rate of $1 \times 10^{-4}$. A cross-entropy loss was utilized for training all models. Each model underwent training for 50 epochs on Utah and 200 epochs on TCGA, utilizing early stopping with patience of 20 and 50 epochs, respectively. We saved the best-performing model from the validation data for inference. Model performances were evaluated using balanced accuracy across all classes for both datasets. All computations were performed on an NVIDIA RTX A6000 with 48 GB of memory. Training our model for 50 epochs on a single gpu takes around 17.4 hours. For data augmentation, we applied color jittering, random rotation, center crop, random crop, and random flips horizontally and vertically for training data. We used the mean and standard deviation from ImageNet to normalize the data. For inference, we used only center cropping and the same normalization.

### 3.2. Result and Discussion

We utilized ResNet18 and ResNet50 backbones (He et al., 2016) to examine the efficacy of our model under two paradigms: fine-tuning with ImageNet supervised pre-training and transfer learning with pathology (The Cancer Genome Atlas-TCGA dataset (Weinstein et al., 2013) and TULIP self-supervised pre-training (Kang et al., 2023). Results, shown in Table 1, demonstrate that our model outperforms the ResNet baselines by over 2% across all settings and exceeds various Hybrid-ViTs in both scenarios. The results underscore our model's capacity to harness multi-scale features and integrate crucial inductive biases without necessitating additional tasks or additional pre-training of the transformer encoder. In the fine-tuning scenario, particularly with TCGA using a ResNet 50 backbone, deeper encoders sometimes hindered performance, highlighting the need for careful design when integrating CNN architectures, especially considering domain shifts. Our DuoFormer improved performance by 3.83%, demonstrating its effectiveness in leveraging multi-scale representations, compared to the shifting window strategy of Swin transformer(Liu et al., 2021b). This also indicates our model's ability to learn representations for the task at hand

| Dataset | Fine-tuning | Params | Acc.(%) |
|---------|-------------|--------|---------|
| TCGA | ViT-Base | 86.57M | $73.50 \pm 0.94$ |
| | ResNet50 | 23.50M | $72.74 \pm 6.22$ |
| | ResNet50-ViT Base | 112.5M | $75.89 \pm 2.60$ |
| | ResNet50-ViT Large | 197.6M | $73.34 \pm 3.72$ |
| | ResNet50-Swin Base | 87.00M | $72.31 \pm 1.68$ |
| | ResNet50-DuoFormer (Ours) | 186.0M | $\mathbf{76.57 \pm 2.23}$ |
| UTAH | ViT-Base | 86.57M | $84.69 \pm 1.33$ |
| | ResNet18 | 11.20M | $88.87 \pm 1.99$ |
| | ResNet18-ViT Base | 99.03M | $82.35 \pm 3.40$ |
| | ResNet18-ViT Large | 184.1M | $86.39 \pm 0.96$ |
| | ResNet18-Swin Base | 86.91M | $84.24 \pm 1.08$ |
| | ResNet18-DuoFormer (Ours) | 91.22M | $\mathbf{91.22 \pm 1.74}$ |

Table 1: Comparison of supervised pretrained models on ImageNet for TCGA and UTAH datasets. Accuracies are reported as mean values from five independent experiments.

and better guide the feature extractor to adapt to domain shifts when trained together. During the transfer learning phase, shown in Table 2, the backbone, self-supervised pretrained on TCGA and TULIP, two large-scale medical datasets, was frozen to serve as a feature extractor. The backbone provided robust visual representations, leading to the most promising performance improvements. Our model significantly outperformed the baseline by 6.96% and clearly surpassed the Hybrid-ViTs and swin transformer, showing the superiority of our model in leveraging multi-scale features. These findings suggest that the model can effectively capture essential local features while preserving global attention capabilities, thereby addressing the typical inductive bias limitations found in transformers.

| Transfer Learning | Params | Accuracy (%) |
|-------------------|--------|--------------|
| SwaV | 0.008M | $77.98 \pm 0.54$ |
| SwaV-ViT Base | 89.03M | $74.00 \pm 1.59$ |
| SwaV-ViT Large | 174.1M | $83.35 \pm 1.90$ |
| SwaV-Swin Base | 86.74M | $68.90 \pm 1.24$ |
| SwaV-DuoFormer(Ours) | 124.7M | $\mathbf{84.94 \pm 2.63}$ |

Table 2: Pathology self-supervised pretrained model performances on TCGA.

### 3.2.1. ABLATION ON MULTI-SCALE REPRESENTATIONS

A natural question to ask is whether it always better to incorporate additional scales, especially in medical datasets characterized by diverse scales. Intuitively, we might expect performance improvements as more scales are integrated. Interestingly, our results reveal significant performance enhancements when utilizing a single scale, which outperforms other baselines and underscores the efficacy of our proposed components. As illustrated in Table 3, model performance generally improves with the incorporation of three and four scales in TCGA, a medium-sized dataset. Conversely, adding more than two stages slightly dimin-

ishes generalization capabilities on the UTAH dataset, given its smaller size. Our findings indicate that the optimal combination varies between datasets, influenced by the dataset size and potentially by the scale of the class-related lesions. Specifically, including scale 1 tends to yield substantial gains, which we attribute to an optimal balance between rich semantic information and manageable embedding lengths.

| Scale 0 | Scale 1 | Scale 2 | Scale 3 | UTAH (Acc. %) | TCGA (Acc. %) |
|---|---|---|---|---|---|
| ✓ | | | | $90.32 \pm 1.61$ | $81.02 \pm 2.04$ |
| | ✓ | | | $90.27 \pm 2.78$ | $81.47 \pm 2.59$ |
| | | ✓ | | $90.10 \pm 0.85$ | $81.13 \pm 1.37$ |
| | | | ✓ | $89.56 \pm 0.53$ | $81.59 \pm 3.54$ |
| ✓ | ✓ | | | $87.91 \pm 1.96$ | $81.64 \pm 1.83$ |
| ✓ | | ✓ | | $86.14 \pm 2.17$ | $80.56 \pm 0.59$ |
| ✓ | | | ✓ | $85.18 \pm 0.63$ | $79.74 \pm 4.35$ |
| | ✓ | ✓ | | $87.35 \pm 1.35$ | $81.36 \pm 2.25$ |
| | ✓ | | ✓ | $\mathbf{91.22 \pm 1.74}$ | $80.07 \pm 1.04$ |
| | | ✓ | ✓ | $90.87 \pm 1.22$ | $80.24 \pm 2.70$ |
| ✓ | ✓ | | ✓ | $89.11 \pm 1.58$ | $82.87 \pm 1.64$ |
| ✓ | ✓ | ✓ | | $87.88 \pm 1.54$ | $81.96 \pm 1.03$ |
| ✓ | | ✓ | ✓ | $88.54 \pm 2.93$ | $81.90 \pm 2.28$ |
| | ✓ | ✓ | ✓ | $89.78 \pm 0.84$ | $84.00 \pm 2.26$ |
| ✓ | ✓ | ✓ | ✓ | $88.59 \pm 1.97$ | $\mathbf{84.94 \pm 2.63}$ |

Table 3: Ablation study on inclusion of scales: Features from different stages are numbered 0 to 3 as in Figure 1 and Figure 2. Mean accuracies from five independent runs are reported.

### 3.2.2. Ablation on Scale Attention

We performed ablations on the local and global attention mechanism in DuoFormer using our optimal models in both transfer learning and fine-tuning settings Table 4. Using only the local attention outperforms setups of replying solely on global attention, which resembles a hybrid ViT model(Dosovitskiy et al., 2020). Moreover, the results demonstrate that optimal performance on both datasets is attained only when both attention modules are combined, emphasizing the necessity of integrating both local and global information.

### 3.2.3. Ablation on Scale Token

The channel dimension embeds rich scale information as it captures different semantic patterns in segmentations(Wang et al., 2022). We experimented with configurations with and without a scale token in our model, as presented in Table 4. Results indicated that our proposed scale token more effectively guides the model in capturing critical local information. For configurations lacking a scale token, we observed enhancements over the baseline model by either using the first token in scale attention or averaging all tokens. Remarkably, employing the first token proved more beneficial than averaging. This token corresponds

to the output from the final stage of the CNN backbone, typically utilized as the input for the classification head. We hypothesize that this performance boost stems from the final stage's ability to convey concise, crucial information for the task, whereas averaging might introduce unwanted noise.

| Method | UTAH | TCGA |
|---|---|---|
| Local Attn | $90.31 \pm 1.15$ | $79.90 \pm 0.10$ |
| Global Attn | $82.35 \pm 3.40$ | $74.00 \pm 1.59$ |
| Ours | $\mathbf{91.22 \pm 1.74}$ | $\mathbf{84.94 \pm 2.63}$ |

| | w/o Scale Token | | w/i Scale Token | |
|---|---|---|---|---|
| Dataset | First Token | Average | Learnable | Ours |
| UTAH | $90.61 \pm 0.69$ | $89.62 \pm 1.40$ | $88.80 \pm 0.78$ | $\mathbf{91.22 \pm 1.74}$ |
| TCGA | $83.22 \pm 1.58$ | $82.62 \pm 0.39$ | $83.13 \pm 0.46$ | $\mathbf{84.94 \pm 2.63}$ |

Table 4: (Top) Ablations on scale and patch attention. Configurations with only scale attention use a single fully-connected layer to adapt the scale token for the classification head. (Bottom) Ablation study on the impact of different scale token configurations.

## 4. Conclusion

We introduced a novel hierarchical transformer model that integrates duo attention mechanisms to enhance visual data interpretation across various scales. Our model effectively captures spatial and contextual information, proving beneficial for medical image classification. Ablation studies confirmed that combining both attention mechanisms optimizes performance, showcasing the model's robustness and versatility across different backbones and tasks. This adaptability paves the way for broader applications in medical imaging and other vision-related challenges.

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

## Appendix A. Ablation on Numbers of Heads and Layers

We assessed our model's sensitivity to two hyperparameters: the number of heads and the number of layers in dual attention modules. Results are given in Table 5. Initially, we fixed the number of heads at 12 and varied the number of layers from 4 to 12 to identify optimal configurations for each dataset. Subsequently, we tested heads from 4 to 12, excluding 10 due to incompatibility with the feature dimension $D = 768$, using the optimal number of layers. We observed that performance generally increases and then decreases with attention depth. Specifically, performance peaks at 6 layers for the Utah dataset and at 8 layers for

the TCGA dataset, likely due to the varying sizes of the datasets. Additionally, we noted a similar pattern of initial increase followed by a decrease in performance for the number of heads across both datasets, peaking at 8 heads.

| Number of Layers | TCGA Acc. (%) | UTAH Acc. (%) |
| --- | --- | --- |
| 4 | 80.83 | 89.37 |
| 6 | 79.70 | 90.41 |
| 8 | 82.67 | 88.64 |
| 10 | 81.09 | 88.87 |
| 12 | 79.66 | 89.86 |

| Number of Heads | TCGA Acc. (%) | UTAH Acc. (%) |
| --- | --- | --- |
| 4 | 78.74 | 90.00 |
| 6 | 82.84 | 90.02 |
| 8 | 84.94 | 91.22 |
| 12 | 82.67 | 90.41 |

Table 5: Ablation studies comparing the impact of different configurations on dual attention modules for both datasets: (a) variations in the number of blocks and (b) variations in the number of heads. All configurations synchronize the blocks and heads in both scale and patch attention. Encoder layers are set to be the optimized ones according to the ablations on blocks.

