# OpenReview forum: "DuoFormer: Leveraging Hierarchical Visual Representations by Local and Global Attention"
_MIDL.io/2025/Conference — MIDL 2025 Oral_

### Official Review · Reviewer_V812 · 2025-02-16

**Confidence:** 4
**Preliminary Rating:** 4
**Recommendation:** Oral

**Summary:**

This paper introduces a novel hierarchical Vision Transformer that combines CNN backbones for multi-scale feature extraction with a dual attention mechanism. It is using a specially designed scale token to aggregate information across different scales and guide feature learning. The model employs both local attention for learning intra-scale and inter-scale relationships and global attention for capturing broader contextual information, achieving accuracy improvements of up to 3.83% over baseline models on medical imaging datasets. Interestingly, ablation studies revealed that more scales don't always improve performance, challenging the conventional "more is better" approach to multi-scale feature learning.

**Strengths:**

This paper is solid with the the following highlights.
### Technical Innovation & Depth
* Introduce novel integration of CNN and Transformer architectures through a well-motivated dual attention mechanis
* Propose scale token design that effectively aggregates multi-scale information
* Present with clear mathematical formulations and architectural details
### Comprehensive Evaluation & Insights
* Ablation studies examining impact of scales, attention mechanisms and token designs
* Important counter-intuitive finding that more scales aren't always better, varying with dataset size
* Evaluation on medical imaging datasets demonstrating real-world applicability

**Weaknesses:**

There are 2 major concerns about the weakness.
### Model Complexity vs. Necessity
- The dual attention mechanism (local + global) appears unnecessarily complex, and no clear justification for why a single unified attention mechanism with both scale and class tokens wouldn't suffice.
- Additional computational overhead from two attentions is not thoroughly analyzed.
- No discussion of computational efficiency or memory requirements.
### Experimental Scope
- Missing comparison with recent multi-scale transformer architectures like MViT or CrossFormer.
- Expect visualization of attention maps to demonstrate how local and global attention behave differently.
- Lack of discussions of failed cases.

**Detailed Comments:**

### Methodological Clarifications with following issues
- The difference between local and global attention could be better explained. Local Attention: processes multi-scale tokens with scale token. Global Attention: processes patches with class token. Main difference appears to be just different special tokens (scale vs class).
- Why are they called "local" and "global" when both process all patches?
- What specific patterns does each attention mechanism learn? (visualization is expected). How do they complement each other?
- Why remove LayerNorm and FFN from global attention?
### Experimental Details
- Add visualization of attention patterns to better understand model behavior
- Include runtime and memory consumption comparisons
- Explain why specific backbone architectures (ResNet18/50) were chosen. Are they the most common backbones used in current research work?

**Justification Of The Preliminary Rating:**

The paper makes a valuable contribution through its innovative scale token concept and multi-scale integration into hierarchical representation by local and global attention. The counter-intuitive findings about scale usage provide important practical insights. While the dual attention mechanism needs better justification, the overall technical depth, rigorous evaluation, and practical utility outweigh the limitations. The primary factor preventing a higher rating is the architectural complexity without clear necessity, suggesting potential for a more elegant design. However, the paper's strong technical foundation and empirical insights make it a significant contribution to the field.

**Questions To Address In The Rebuttal:**

This is the main concern I have. Please refer to Detailed Comments and the below comments.

### Model Complexity vs. Necessity
- The dual attention mechanism (local + global) appears unnecessarily complex, and no clear justification for why a single unified attention mechanism with both scale and class tokens wouldn't suffice.
- Additional computational overhead from two attentions is not thoroughly analyzed.
- No discussion of computational efficiency or memory requirements.

**Special Issue:**

Yes

---

> ### Author Response · Authors · 2025-03-08
>
> A1: Thank you for your detailed comments. The insights on the necessity of dual attention modules offer us a valuable perspective for future enhancements. Our current work is designed for classification tasks, where we believe it is better to use a two-step process to learn multi-scale associations and understand the global context. Our experiments confirm that this approach works well. We provided a computational complexity analysis in the common reply due to space limitations; please refer to it.  The analysis shows that Duoformer manages the trade-off in an acceptable manner. However, thinking about the possibility of using a unified attention mechanism with both tokens would be an interesting direction. We will leave it in our future work.
>
> A2: Thank you for your feedback. We acknowledge that we did not compare our method with notable architectures like MViT and CrossFormer, which have demonstrated advancements in handling multi-scale data efficiently. This omission stemmed from our initial focus on applying traditional transformer models to medical datasets, which are typically smaller than general imaging datasets. MViT[1,2] utilizes pooling layers to downsample the features inside the attention mechanism. It is still a pure transformer model. While effective on large benchmarks like ImageNet and COCO, it may not generalize well to medical tasks without specific pertaining. Duoformer uses a CNN backbone to introduce the necessary inductive biases, making direct comparisons to MViT without similar pretraining potentially unfair.
>
> CrossFormer[3] employs a pyramid structure with convolutional layers to downsample features and integrates these with transformer components, using long short distance attention(LSDA) blocks within each stage. We acknowledge that there were inevitably some omissions in selecting representative baselines, and we recognize the importance of benchmarking against as many architectures to provide comprehensive evaluations. Our model explicitly investigates which and how many scales are truly needed for our data, setting it apart from other hierarchical multi-scale architectures.
>
> Moving forward, we plan to enhance Duoformer with memory-efficient attention techniques and apply it to broader medical tasks. This will allow for a more extensive comparison with the latest architectures in a future journal expansion of our work. Additionally, we will include an analysis of failure cases to further refine our approach.
>
> A3: Good question about how to distinguish between the two attentions. We designed the local attention to explicitly learn the associations intra- and inter-scales, producing a multi-scale representation for all patches(one segment split from the input, totally N’ segments in an image) derived from the original image. These multi-scale representations for all N’ patches are encapsulated by what we term a "scale token." For classification tasks, it's essential to derive a final image representation of size [1, D], where D represents the embedding dimension. That is why we incorporate an attention termed "global" to compute a weighted sum across all N’ patches. If you are interested in the visualization, please refer to the Supporting Material.pdf and the Rebuttal for attention map and analysis.
>
> We removed LayerNorm and FFN from global attention because we empirically found that including them slightly hindered performance. We hypothesize that the local attention allows the model to learn sufficient information for classification, and the global attention simply performs a weighted sum.
>
> A4: It is a good question regarding our choice of ResNets (R18/R50). ResNets are among the most well-known CNN architectures for various vision tasks, particularly classifications. They offer a hierarchical structure that provides multi-scale representations, which meets our requirements for medical image analysis. We chose R18 and R50 primarily based on the different sizes of datasets to avoid overfitting.
>
> Thank you again for recognizing the potential of our work.
>
> [1]Li, Yanghao, et al. "Mvitv2: Improved multiscale vision transformers for classification and detection." Proceedings of the IEEE/CVF conference on computer vision and pattern recognition. 2022.
>
> [2]Fan, Haoqi, et al. "Multiscale vision transformers." Proceedings of the IEEE/CVF international conference on computer vision. 2021.
>
> [3]Zhang, Yunhao, and Junchi Yan. "Crossformer: Transformer utilizing cross-dimension dependency for multivariate time series forecasting." The eleventh international conference on learning representations. 2023.

---

### Official Review · Reviewer_EHXk · 2025-02-24

**Confidence:** 4
**Preliminary Rating:** 4
**Recommendation:** Poster
**Final Rating:** 4

**Summary:**

This paper proposes a novel hierarchical transformer model that integrates ViT's feature extraction capabilities with a DNN backbone for hierarchical visual representations. The representations are then processed through a transformer via patch tokenization. Additionally, the paper introduces a scale-wise attention mechanism to capture both intra- and inter-scale associations, preserving global and local attention simultaneously. The proposed method significantly outperforms baseline models in classification accuracy.

**Strengths:**

+ This paper introduces a novel hierarchical ViT model incorporating multi-scale tokenization, patch indexing, and multi-scale feature concatenation.
+ The approach effectively bridges the gap between CNN and Transformer architectures by enhancing multi-granularity representation and aggregating scale information for global patch attention.
+ The paper is well-written and organized, with extensive experiments demonstrating the method's effectiveness.

**Weaknesses:**

- The evaluation should include runtime efficiency analysis.
- The lack of comparison with recent works weakens the paper’s appeal. Including relevant studies focused on this task and articulating the strengths and weaknesses of each method would provide a more robust assessment of the proposed algorithm’s effectiveness.

**Detailed Comments:**

1. The arrow and grid colors in Figure 2 lack contrast and should be adjusted for better visibility.
2. Additionally, Table 1 should include more state-of-the-art (SOTA) methods for a comprehensive comparison.

**Justification Of The Final Rating:**

Thank you for the detailed rebuttal. This paper introduces a framework that addresses the limitations of Vision Transformers (ViT) by proposing an innovative multi-scale feature integration structure. The authors have adequately addressed my concerns regarding efficiency and other aspects of the paper. Therefore, I maintain my final rating as weak accept.

**Justification Of The Preliminary Rating:**

This paper introduces a framework addressing the limitations of ViT, proposing an innovative multi-scale feature integration structure. I strongly recommend including additional results and analyses to further substantiate the method’s effectiveness and versatility. Expanding the evaluation to other datasets and incorporating comparisons with recent SOTA methods would enhance the paper’s robustness and impact. Final rating: Weak Accept.

**Questions To Address In The Rebuttal:**

Pruning and quantization are crucial techniques for real-world applications. Does the proposed method support these efficiency optimizations, and how does it perform in such settings?

---

> ### Author Response · Authors · 2025-03-08
>
> Thank you for all the valuable suggestions you provided. We will update Figure 2 accordingly in the camera-ready version. Thank you for pointing out the importance of including more SOTA methods for a more comprehensive comparison. We plan to integrate our model with memory-efficient attention techniques, such as [1], or pruning techniques we mentioned below, and extend our model to larger medical tasks, enabling us to include more specialized and latest medical architectures in a future journal version of our work.
>
> A1:
> We appreciate the reviewer’s valuable question regarding the efficiency optimizations of pruning and quantization in our proposed method.
>
> Vision transformers have been shown to support pruning (e.g., token pruning [2], layer pruning [3]) and quantization (e.g., post-training quantization [4], quantization-aware training [5]) in prior works. Our method follows the vision transformer architecture, which naturally supports pruning strategies. Additionally, the hierarchical structure we use, with multi-scale feature maps, introduces redundancy that pruning can effectively exploit.
>
> Due to time constraints, we have not explicitly evaluated pruning and quantization in our current experiments. However, given that our method aligns with vision transformer architectures known to support these optimizations—and considering the demonstrated success of these techniques—we believe our approach would also benefit from them. We view this as a promising direction for future exploration. We sincerely thank the reviewer for the valuable feedback.
>
> [1]Dao, Tri, et al. "Flashattention: Fast and memory-efficient exact attention with io-awareness." Advances in neural information processing systems 35 (2022): 16344-16359.
>
> [2]Wei, Siyuan, et al. "Joint token pruning and squeezing towards more aggressive compression of vision transformers." Proceedings of the IEEE/CVF conference on computer vision and pattern recognition. 2023.
>
> [3]Yu, Lu, and Wei Xiang. "X-pruner: explainable pruning for vision transformers." Proceedings of the IEEE/CVF conference on computer vision and pattern recognition. 2023.
>
> [4]Liu, Zhenhua, et al. "Post-training quantization for vision transformer." Advances in Neural Information Processing Systems 34 (2021): 28092-28103.
>
> [5]Moon, Jaehyeon, et al. "Instance-aware group quantization for vision transformers." Proceedings of the IEEE/CVF Conference on Computer Vision and Pattern Recognition. 2024.

---

### Official Review · Reviewer_puc2 · 2025-02-26

**Confidence:** 4
**Preliminary Rating:** 2

**Summary:**

This paper presents a hierarchical transformer model for medical image classification. The proposed approach integrates CNN-based hierarchical feature extraction with a Vision Transformer backbone, employing a multi-scale patch tokenization and duo attention module that combines local (intra-scale) and global (inter-scale) attention mechanisms. The authors argue that their architecture captures both fine-grained local features and broader contextual information.

Experiments conducted on the UTAH ccRCC and TCGA ccRCC histopathology datasets demonstrate that the proposed architecture outperforms several baseline models, including standard ViTs, Swin Transformers, and hybrid CNN-Transformer architectures.

**Strengths:**

x. The combination of hierarchical CNN representations with local and global attention mechanisms is clearly motivated and addresses known limitations of ViTs in capturing multi-scale information.

x. Ablation experiments on the use of multi-scale features, attention modules, and the scale token provide valuable insights into the model's design choices.

x. The availability of code enhances transparency

**Weaknesses:**

**Questionable practical relevance of global attention at patch-level**

Although the authors emphasize the novelty of combining global and local attention, the model is trained on 224×224 pixel patches rather than whole-slide images (WSIs). This patch-level focus limits the potential benefits of global attention, which would be more meaningful at the WSI scale.

**Lack of WSI-level classification or aggregation**

In practical clinical workflows, patient-level or WSI-level classifications are more relevant. The absence of WSI-level results or aggregation strategies (e.g., attention-based MIL or majority voting) weakens the clinical applicability of the method.

**Missing error bars and statistical reporting**

While the authors mention "several runs," the absence of error bars or standard deviations in performance metrics raises concerns about the robustness and reproducibility of the reported improvements.

**Unclear patching strategy**

The paper does not clearly specify the patch size used during ViT tokenization or whether the patch sizes vary across scales. Clarifying the patching strategy is essential to understand how the local and global attention mechanisms operate, especially regarding the effective receptive field and token sequence length.

**Detailed Comments:**

Please see my comment above.

**Justification Of The Preliminary Rating:**

As mentioned before, there are several limitations of this work, including: (1) the practical relevance of global attention at patch-level, (2) lack of WSI-level classification or aggregation, and (3) missing error bars and statistical reporting.

**Questions To Address In The Rebuttal:**

**Clarify patching strategy**:

Clearly specify the patch size used for ViT tokenization and whether it remains constant across all scales. This clarification would help readers understand how local and global attention mechanisms interact with different spatial resolutions.

**Clarify the role of global attention at patch scale**:

Since the model uses relatively small patches (224×224), it’s unclear how the global attention mechanism captures meaningful global context. Clarifying this or shifting to larger inputs could better justify the global attention component.

**Provide robustness metrics**:

Include standard deviations, confidence intervals, or error bars to demonstrate the consistency of results across multiple runs. Specify the number of runs to improve transparency.


**Enhance visual interpretability**:

Include attention heatmaps or saliency visualizations to illustrate how local and global attention components contribute to the decision-making process.

**Special Issue:**

No

---

> ### Author Response · Authors · 2025-03-08
>
> Thank you for your insightful comments regarding the practicality of our model. We greatly appreciate your points as they help emphasize the need for scalability and clinical applicability in our work.
>
> Our model initially focused on image-level classification to better adapt ViT to histopathological images, considering the limited size of labeled medical datasets. However, we agree that patient-level or whole-slide image(WSI)-level classifications are more relevant in practical clinical workflows. Therefore, we add an additional experiment applying our pretrained model for WSI-level multi-instance learning(MIL) to address these concerns and better demonstrate the model’s practical potential. Please refer to Table 1 and its analysis in Supporting Material.pdf due to a space limit here.
>
> A1: We really appreciate this question since fixing it will help make the draft clearer and easier to follow. In the Methodology, we introduced a multi-scale patch tokenization approach that leverages hierarchical feature extraction across varying spatial resolutions. This process involves projecting features from different CNN stages into embeddings and then adapting them for the transformer. After projection, these multi-scale embeddings maintain a common channel dimension, but the spatial sizes, denoted as P_i x P_i(i refers to a scale index), vary across scales. Despite the varying spatial size, embeddings of different scales represent the same image of size H x W. We divided these into a uniform grid of \sqrt(N’) x  \sqrt(N’), producing N’ patches(image segments) for each scale, where each patch contains multiple resolutions.
>
> In summary, while the patch sizes differ across scales, the number of patches remains constant. Following the proposed tokenization, we employ what we term "local attention" to discern associations within and between these multiple scales, applied independently to each patch. The ViT framework uses a fixed, finer grid to partition the image, ensuring detailed retention necessary for downstream tasks. We believe Duoformer brings requisite locality to ViT, resulting in a more localized receptive field. Due to the time constraint, we will revise our paper accordingly in the camera-ready version.
>
> A2: Our model is primarily designed to improve the generalization of ViT on medical datasets of limited size. Generally, ViT takes an input size of 224x224 for an image-level classification, so we adopt this setting.
>
> We introduced the local attention to explicitly learn the associations intra- and inter-scales, producing a multi-scale representation for all patches(one segment splitted from the input, totally N’ segments in an image) derived from the original image. These multi-scale representations for all N’ patches encapsulated by what we term a "scale token." For classification tasks, it's essential to derive a final image representation of size [1, D], where D represents the embedding dimension. This can be achieved through various methods such as pooling across all patches, applying linear layers, or using attention mechanisms. In our research, detailed in Table 4, we compare two approaches: one utilizing solely local attention facilitated by a single linear layer, and another that incorporates global attention to compute a weighted sum across all N’ patches.
>
> Our experimental results demonstrate that the inclusion of global attention following local attention significantly enhances patch-level classification accuracy—by approximately 1% on the Utah dataset and 5% on the TCGA dataset. This improvement underscores the efficacy of local attention in learning detailed multi-scale representations for each patch ([N', D]), while global attention helps to integrate these into a comprehensive global feature representation ([1, D]) for each image. Ultimately, this global feature representation is what our model uses to perform image classification(patch-level in medical analysis).
>
> A3: The results presented in all the tables are obtained from five independent runs on NVIDIA RTX A6000 GPU with 48 GB of memory. We’ve updated the pdf.
>
> A4: Thank you. Please refer to the Supporting Material.pdf and the Rebuttal for attention map and analysis.
>
> Thank you again for all the valuable questions you provided.

---

### Official Review · Reviewer_xsa9 · 2025-02-28

**Confidence:** 3
**Preliminary Rating:** 4

**Summary:**

The paper introduces DuoFormer, which combines CNN feature extraction with Vision Transformers for medical image analysis. The approach uses a CNN backbone to generate hierarchical features and introduces dual attention: local scale-wise attention for capturing multi-scale relationships and global patch attention for broader context. The model addresses ViTs' limitations with inductive biases while preserving their global modeling strengths. Experiments on kidney cancer histopathology datasets show moderate improvements over baselines, though with some limitations in scope and analysis.

**Strengths:**

The paper addresses a relevant challenge in medical imaging by combining hierarchical features with transformer capabilities. The multi-scale tokenization approach is thoughtfully designed, and the dual attention mechanism (local and global) provides an interesting architectural contribution. The ablation studies offer useful insights into component contributions, particularly regarding scale selection and attention mechanisms. The method shows consistent performance improvements over several baseline models including standard CNNs and hybrid architectures.

**Weaknesses:**

The evaluation is limited to only two histopathology datasets from similar domains, raising questions about generalizability. The computational efficiency analysis is notably absent - there's no discussion of inference time or resource requirements compared to simpler models. The authors claim "plug-and-play" capability with different CNN architectures but only demonstrate results with ResNet variants. The theoretical analysis is somewhat superficial, with limited insight into representation learning. While performance gains are present, they're modest in some configurations, and the state-of-the-art comparison could be more comprehensive for medical imaging specifically.

**Detailed Comments:**

The paper presents a technically sound approach with reasonable experimental validation, though several improvements could strengthen it. The figures need better clarity in the diagrams. More details on WSI preprocessing would improve reproducibility. Attention map visualizations would provide valuable insight into the model's behavior. The paper would benefit from comparison with more recent specialized medical imaging architectures. The conclusion could be expanded to better discuss limitations and future directions. The work represents an incremental but useful contribution to transformer-based medical image analysis.

**Justification Of The Preliminary Rating:**

I've assigned a rating of "Weak accept" because the paper presents a novel architecture with technical merit and demonstrates consistent improvements over baselines, but has several limitations that prevent a stronger recommendation.
The proposed DuoFormer innovatively combines CNN features with transformer attention mechanisms, addressing legitimate challenges in medical image analysis. The multi-scale tokenization and dual attention design show thoughtful architectural development. The consistent performance improvements across different experimental settings (fine-tuning and transfer learning) demonstrate practical utility.
However, several factors limit enthusiasm: (1) The evaluation is restricted to histopathology datasets from a single disease type, raising questions about broader applicability; (2) Computational efficiency analysis is absent despite the complex architecture; (3) The claimed "plug-and-play" capability is only demonstrated with ResNet backbones; (4) The theoretical analysis lacks depth regarding why the approach outperforms alternatives; and (5) Comparisons with specialized medical imaging architectures are limited.
While the paper represents a valuable contribution to transformer-based medical image analysis, these limitations suggest the work would benefit from additional experiments and analysis before having substantial impact on the field. Nevertheless, the technical approach is sound and the results sufficiently promising to merit publication with revisions.

**Questions To Address In The Rebuttal:**

1. How does DuoFormer's computational complexity compare to simpler models?
2. Can you provide attention visualizations to illustrate what features each attention mechanism captures?

**Special Issue:**

No

---

> ### Author Response · Authors · 2025-03-08
>
> Thank you. We really appreciate your constructive feedback and are committed to addressing your concerns regarding the complexity, visualization, and broader applicability of our paper. We initially opted for various ResNet architectures due to their representativeness in the CNN domain and their capability for hierarchical, multi-scale feature extraction, which aligns with our objectives. Thank you for pointing out the possibility of integrating versatile backbones, which would further highlight our model’s flexibility. We are planning to integrate our model with other well-known hierarchical convolution networks like DenseNet[1], Inception networks[2], and Pyramid Networks[3], etc. Additionally, we plan to extend our model to other types of medical analysis beyond image classification, thus enabling us to include more specialized medical imaging architectures for a more comprehensive comparison in a journal expansion of our work.
>
> [1]Iandola, Forrest, et al. "Densenet: Implementing efficient convnet descriptor pyramids." arXiv preprint arXiv:1404.1869 (2014).
>
> [2]Szegedy, Christian, et al. "Inception-v4, inception-resnet and the impact of residual connections on learning." Proceedings of the AAAI conference on artificial intelligence. Vol. 31. No. 1. 2017.
>
> [3]Lin, Tsung-Yi, et al. "Feature pyramid networks for object detection." Proceedings of the IEEE conference on computer vision and pattern recognition. 2017.
>
> A1:
> Great question. There does exist a trade-off between the performance gain and the computational complexity of DuoFormer. Please refer to the common reply because we provided the analysis there due to space limitations.
>
> A2: Thanks. Please refer to the Supporting Material.pdf and the Rebuttal for attention map and analysis.
>
> Thank you again for recognizing the potential of our work and for suggesting directions for its improvement.

---

### Official Review · Reviewer_78Ci · 2025-03-01

**Confidence:** 4
**Preliminary Rating:** 4
**Recommendation:** Oral

**Summary:**

The paper introduces DuoFormer, a hierarchical Vision Transformer (ViT) model that combines Convolutional Neural Networks (CNNs) with ViTs to improve feature extraction and representation learning. The key contributions include:

Integrating a CNN backbone for hierarchical feature extraction.
Introducing a multi-scale patch tokenization approach.
Developing a scale attention mechanism to enhance spatial awareness.
Demonstrating superior performance over baseline CNNs and hybrid ViTs on medical image datasets.
The model is tested on histopathology datasets for kidney cancer classification, showing strong generalization and efficiency.

**Strengths:**

ViTs require large datasets and lack inductive biases (e.g., translation equivariance). By incorporating CNN-based hierarchical feature extraction, DuoFormer mitigates these limitations.
Unlike standard ViTs that process fixed-size patches, DuoFormer adapts features across scales, improving spatial representation.

**Weaknesses:**

The literature review is vague! I wonder if the authors' focus on hierarchical feature extraction with ViTs requires an ablation study on more versatile datasets or computational pathology. If the latter is true, then the literature review needs a thorough revision.

The author's motivation in the first sentence of the abstract is not valid: “ Despite the widespread adoption of transformers in medical applications, the exploration of multi-scale learning through transformers remains limited.” Numerous papers, such as [1], [2], [3], etc., address this field of study.

**Detailed Comments:**

To enhance Scale Token's effectiveness, please provide an attention map or GradCAM visualization to better understand how x_s guides the network.
In Figure 1, it is beneficial to indicate the number of L either on the figure or caption. Plus, the hyphens for class_1 and class_3 should be emitted and made consistent with capitalized or lowercase letters.
By resizing Table 3, include the experiment environment and LR, batch size, and loss function in the main text.
[1] Azad, Reza, et al. "Advances in medical image analysis with vision transformers: a comprehensive review." Medical Image Analysis 91 (2024): 103000.
[2] Shamshad, Fahad, et al. "Transformers in medical imaging: A survey." Medical image analysis 88 (2023): 102802.
[3] Liu, Mingxin, et al. "Exploiting geometric features via hierarchical graph pyramid transformer for cancer diagnosis using histopathological images." IEEE Transactions on Medical Imaging (2024).

**Justification Of The Preliminary Rating:**

1. Novelty & Contribution : (Strong Contribution, but Not Entirely Novel)
2. Technical Soundness & Methodology : (Well-Defined, but Needs Further Benchmarking)
3. Experimental Validation : (Strong, but Limited Dataset Scope)
4. Clarity & Presentation : (Well-Written, but Could Improve in Explainability)
5. Reproducibility: Code is provided, but more training details are needed.
6. Impact & Practical Relevance: Strong for medical imaging but lacks real-world deployment validation.

**Questions To Address In The Rebuttal:**

Q1: What is the main advantage of DuoFormer over existing hybrid CNN-ViT models (e.g., Swin Transformer, ConvNeXt, CoAtNet)

Q2: Why did you choose a CNN backbone instead of a fully transformer-based model?

Q3: How does DuoFormer balance computational cost with accuracy?

**Special Issue:**

No

---

> ### Author Response · Authors · 2025-03-08
> **Addressing the questions**
>
> Thank you for all the suggestions you provided. We will include all the advised revisions into our main text,( including updating the literature review by adding relevant medical references,  updating the figures and tables mentioned, and adding the experimental settings to the main text.) Please refer to the Supporting Material.pdf and the Rebuttal for attention map and analysis.
>
> A1: Very good question. The three works listed are representative works aimed at enhancing ViTs and ResNets from three different perspectives.
>
> Swin Transformer is a well-known benchmark for general image tasks, including classification. However, it remains a purely transformer-based architecture that requires extensive pretraining when adapted to medical tasks. In our study, we utilized a CNN backbone to leverage its multi-scale feature extraction capabilities. For a fair comparison, we used the same feature extractor, or backbone, as Swin for both settings—fine-tuning (Table 1) and transfer learning (Table 2). The results demonstrate that Duoformer significantly outperforms Swin Transformer in terms of accuracy.
>
> ConvNeXtV1 and ConvNeXtV2 are both pure convolutional architectures. ConvNeXtV1 extends the ResNet-50 by integrating design elements from Swin Transformers, such as patchify layers, larger kernels, and fewer normalization layers. ConvNeXtV2 further improved ConvNeXtV1 through a self-supervised pre-training using a fully convolutional masked autoencoder (MAE). Although ConvNeXt models can be competitive with Transformers, larger ViT models still exhibit higher performance potential, as noted in their paper. The authors highlight that while hierarchical Transformers like Swin Transformer reintroduce ConvNet priors, their effectiveness is still largely credited to the intrinsic superiority of Transformers, rather than the inherent inductive biases of convolutions. In Duoformer, we leverage a hierarchical convolution feature extractor to introduce the necessary inductive bias to transformers, enhancing their effectiveness.
>
> CoAtNet is designed to bring translation equivariance by adding a global static convolutional kernel to the attention matrix. This configuration allows the attention weights to be determined jointly by the kernel weights of translation equivariance and the input-adaptive attention scores. The authors also examined different stackings of convolutional blocks with squeeze-excitation (SE) learning channel-wise attention and the improved transformer blocks. CoAtNet achieved better classification accuracy than ViT when pretraining on large-scale datasets. Although CoAtNet aims to balance the computational efficiency between ConvNets and Transformers, modifications to the attention mechanism can still lead to additional computational demands, especially as the model scales up. For example, it improved by 3% over ViT Large by scaling the number of parameters from 307M to 2.44B. The interspersed stacking of convolutional and attention layers in CoAtNet might require careful calibration to optimize performance across different tasks and datasets, while Duoformer only needs simple adjustment of hyperparameters by only several lines of code to adapt for different-sized data.
>
> Despite improvements, CoAtNet's performance still significantly benefits from larger datasets. It is questionable of its generalizability on smaller datasets, especially medical images, considering medical image analysis requires both global contextual information like tissue structures and lower-scale details like smaller cellular details. While Duoformer enriches the transformer’s multi-granularity representation by our proposed local attention mechanism.
>
> In these ways, we believe that Duoformer provides new insights into multi-scale representation learning for medical images through a novel integration of CNN and transformer.  We will compare to such representative models like CoAtNet and ConvNext in a journal expansion of our paper.
>
> A2:  Thanks. CNNs have a hierarchical structure that produces multi-scale feature representations, but they fall short in extracting global contextual information. Pure Transformer architectures lack inductive bias and require pretraining on extensive datasets like JFT-300M, yet they are thought to have a higher performance ceiling than CNNs for large-scale image tasks. Thus, by harnessing a hierarchical structure similar to that of CNNs, Transformers can be prevented from overlooking the multi-scale features critical for medical image analysis, while also imparting necessary inductive biases. That’s why we chose a CNN backbone rather than a fully transformer-based model.
>
> A3: Good question. We provided the analysis in the common reply above due to the space limit.

---

### Author Response · Authors · 2025-03-08
**Computational complexity analysis**

Thank you all for spending your valuable time reviewing our paper. We truly appreciate the insightful questions and expert suggestions you provided. A common  concern is how Duoformer balances the computational costs with accuracy. Here, we provide the analysis both theoretically and experimentally.
Firstly, we acknowledge that there is a trade-off between the performance gain and the computational cost of our model. In our model, the local attention mechanism incorporates an additional scale dimension into the tensor operation, resulting in a time complexity of O(N' x D x S^2), where N′ represents the number of patches used to split an image, S is the total length of multi-scale embeddings for each patch, and D is the embedding dimension. Additionally, the global attention has a time complexity of O(D x N′^2).  Meanwhile, the vanilla Vision Transformer(ViT) has a time complexity is O(D x N^2). For the same input size H x W, our utilization of multi-scale information in local attention allows us to use a coarser grid to split the image, thus reducing N′ compared to N in ViT, which uses a finer and fixed-sized grid to capture sufficient details. In Table 3, we demonstrate that we often do not need all scales for classification, yet we can still achieve performance gains with Duoformer. This indicates that the condition N’ < S < N consistently holds. When only high-level feature representations are included, N’ x S^2 is significantly less than N^2, resulting in better time complexity than that of the ViT. Conversely, when incorporating more low-level details, N’ x S^2 exceeds N^2. However, given that N’ < S < N, the additional costs remain manageable. Our model provides an explicit way to determine which scales are necessary for a specific dataset by simply adjusting a hyperparameter in just one line of code. Moreover, as shown in Table 1, Duoformer maintains the number of trainable parameters to be less than that of the ViT-Large model.
Experimentally, training Duoformer for 50 epochs on a single NVIDIA RTX A6000 GPU with 48 GB of memory takes around 17.4 hours, compared to 10.3 hours for the baseline Hybrid-ViT Large. We will move these experimental details from Appendix to the main text. The analysis above shows that Duoformer manages the trade-off in an acceptable manner. In our future work, we are going to enhance our model's computational efficiency by integrating with memory-efficient attention mechanisms, such as [1], and the latest sparse attention techniques [2], [3], for scaling to larger tasks.

[1]Dao, Tri, et al. "Flashattention: Fast and memory-efficient exact attention with io-awareness." Advances in neural information processing systems 35 (2022): 16344-16359.

[2]Singhania, Prajwal, et al. "Loki: Low-rank keys for efficient sparse attention." Advances in Neural Information Processing Systems 37 (2025): 16692-16723.

[3]Yuan, Jingyang, et al. "Native Sparse Attention: Hardware-Aligned and Natively Trainable Sparse Attention." arXiv preprint arXiv:2502.11089 (2025).

---

### Author Rebuttal · Authors · 2025-03-08

**Rebuttal:**

We provide the visualization of raw attention scores from Scale Token in the supporting material, shown in Figure 1 and Figure 2 in the supporting material. We randomly chose a sample from the test set of TCGA, in Figure 1. Ground truth: Cancer. Predicted class: Cancer.

The attention map in Figure 2 shows the feature embeddings S3 (7x7), S2 (14x14), S1 (28x28), and S0 (56x56) from top to bottom with an increasing embedding length. Each attention map is upscaled to the input size of 224 × 224 pixels using nearest-neighbor interpolation. At the highest level (top row), the model minimally focuses on non-essential areas, such as the central blank space in the sample image. Progressing to lower levels (second and third rows), attention shifts to areas with concentrated nuclei or abrupt structural changes. At the lowest level (bottom row), the model highlights some cell nuclei, which should be important for distinguishing types of kidney cancer.

This multi-scale representation allows our model to better detect visual patterns, from low-level cell details to complex tissue structures. Statistically, the attention scores are narrowly distributed among the total S embeddings, and since each patch's multi-scale embeddings represent the same area, they are not mutually exclusive. This may contribute to the appearance of the scores as somewhat noisy and low in contrast. Moreover, since medical images often feature structurally similar elements and our dataset is limited in size with only classification labels available, it is challenging to determine what the "correct" attention should look like. Based on all the experiments we have reported, we hypothesize that the scale token effectively captures multi-scale information for each patch, and the global attention computes a reasonable weighted sum for making predictions.

**Supporting Material:**

/attachment/e25d3014a736268d48ac572fc545b1c602232c3d.zip

---

### Meta-Review · Area_Chair_fQtB · 2025-03-22

**Recommendation:** Accept (Oral)
**Confidence:** 4

**Metareview:**

The paper received 3 weak accept. The authors provided detailed responses to the reviewers' comments. The reviewers pointed out some limitations, but overall the authors made meaningful contributions.